# RapidResa Polymyxin *Acinetobacter* NP^®^ Test for Rapid Detection of Polymyxin Resistance in *Acinetobacter baumannii*

**DOI:** 10.3390/antibiotics10050558

**Published:** 2021-05-11

**Authors:** Maxime Bouvier, Mustafa Sadek, Stefano Pomponio, Fernando D’Emidio, Laurent Poirel, Patrice Nordmann

**Affiliations:** 1Medical and Molecular Microbiology, Faculty of Science and Medicine, University of Fribourg, 1700 Fribourg, Switzerland; maxime.bouvier@unifr.ch (M.B.); mustafa.sadek@unifr.ch (M.S.); laurent.poirel@unifr.ch (L.P.); 2Swiss National Reference Center for Emerging Antibiotic Resistance (NARA), University of Fribourg, 1700 Fribourg, Switzerland; 3Liofilchem, 64026 Roseto degli Abruzzi, Italy; spomponio@liofilchem.com (S.P.); fdemidio@liofilchem.com (F.D.); 4INSERM European Unit (IAME), University of Fribourg, 1700 Fribourg, Switzerland; 5Institute for Microbiology, University of Lausanne, 1015 Lausanne, Switzerland; 6Lausanne University Hospital, 1011 Lausanne, Switzerland

**Keywords:** colistin, *Acinetobacter*, susceptibility testing, diagnostic test

## Abstract

A homemade and culture-based test, relying on the visual detection of the reduction of the resazurin reagent (a cell viability indicator), has been developed for the rapid detection of polymyxin resistance in *Acinetobacter baumannii*. Here, we evaluated the industrial version of this test, the RapidResa Polymyxin *Acinetobacter* NP^®^ test (Liofilchem, Italy). A well-characterized panel of 68 clinical *A. baumannii* strains (36 polymyxin-susceptible, 26 polymyxin-resistant *A. baumannii*, and 6 colistin-heteroresistant isolates) of worldwide origin was tested. All the colistin-susceptible *A. baumannii* isolates gave negative results according to the RapidResa Polymyxin *Acinetobacter* NP^®^ test, except for a single isolate that gave a false-positive result. Out of the 26 colistin-resistant *A. baumannii* strains, 25 were correctly identified as colistin resistant using the RapidResa Polymyxin *Acinetobacter* NP^®^ test. Only a single colistin-resistant *A. baumannii* strain gave a false-negative result. Additionally, the six colistin-heteroresistant isolates tested gave positive results. Altogether, the sensitivity and the specificity of the test were found to be 96% and 97%, respectively. The turn-around-time of this easy-to-perform test was 3-4h, which showed excellent reliability for identification of polymyxin resistance in *A. baumannii*. The RapidResa Polymyxin *Acinetobacter* NP^®^ test allows a rapid differentiation between polymyxin-susceptible and -resistant *A. baumannii* isolates, which may contribute to optimization of the use of polymyxins for treating infections due to multidrug-resistant *A. baumannii.*

## 1. Introduction

Multidrug resistance in *Acinetobacter baumannii* is becoming a source of major concern worldwide [1]. This opportunistic pathogen is intrinsically resistant to many antibiotic agents and can also acquire resistance to all antibiotics, virtually. Life-threatening nosocomial infections associated with these multidrug-resistant (MDR) species are increasingly reported, particularly in patients admitted to intensive care units, and may lead to increased mortality rates due to limited therapeutic options [2,3,4]. Very high percentages of carbapenem-resistant *A. baumannii* have been observed worldwide during the past decade [1,5]. Therefore, polymyxins, such as colistin, are now considered as last resort antibiotics for the treatment of infections with carbapenem-resistant *A. baumannii* (CRAB) [6,7,8]. Currently, colistin resistance is a serious global concern due to limited alternative antibiotics [9,10]. Owing to the global increasing occurrence of colistin resistance and heteroresistance in this species, the rapid detection of colistin susceptibility is critical to optimally adapt the empirical treatment [11]. Choice of a polymyxin-based treatment is mainly based on the result of susceptibility testing performed through different techniques. The antimicrobial susceptibility testing for colistin is technically challenging [12,13]. The current standard method of detection for polymyxin susceptibility in Gram negatives is the determination of minimum inhibitory concentration (MIC) by the broth dilution (BMD) method (https://eucast.org/clinical_breakpoints/ accessed on 6 January 2021). However, this procedure is time consuming (24 h), and not easy to implement for most clinical laboratories [14]. 

Recently, the Rapid ResaPolymyxin *Acinetobacter* NP test has been developed [15]. This culture-based technique allows categorization between colistin-susceptible/-resistant *A. baumannii* isolates in 4 h. It is based on the utilization of resazurin (7-hydroxy-3H-phenoxazin-3 one 10-oxide), also referred to as the alamarBlue^®^ and the prestoBlue^®^, by growing bacteria. Its principle is based on the fact that metabolically active cells reduce the blue resazurin to the pink product resorufin. This reduction is proportional to the number of metabolically active cells. Detection of the blue/pink color of the wells is made visually and does not require specialized equipment. Bacterial viability of *A. baumannii* isolates, after growth in medium with or without a defined concentration of colistin, is therefore evidenced [15]. The main value of this test is rapid categorization of susceptibility and resistance, which is important to optimally adapt the empirical treatment [16,17]. 

A commercial version of this test, the RapidResa Polymyxin *Acinetobacter* NP^®^ test (Liofilchem, Roseto degli Abruzzi, Italy), is now available, based on the same protocol as its homemade version. This test complements the Rapid Polymyxin NP^®^ test, which works with Enterobacterales but is not appropriate for non-fermenters, such as *A. baumannii* [18]. The aim of our present study was to evaluate this novel commercial test using a collection of polymyxin-susceptible and polymyxin-resistant *A. baumannii* isolates by comparison with the gold standard method for colistin susceptibility testing and to determine the sensitivity and the specificity of this commercial test.

## 2. Materials and Methods 

A well-characterized panel of 68 clinical *A. baumannii* strains (36 polymyxin-susceptible, 26 polymyxin-resistant *A. baumannii*, and 6 colistin-heteroresistant isolates) of worldwide origin (either from colonization or true infections) was used in this study. The colistin-resistant *A. baumannii* R3402 strain (positive control) and the colistin-susceptible *A. baumannii* R4 strain (negative control) were used. The polymyxin susceptibility or resistance of the strains was determined by the precise determination of MIC values using the broth dilution (BMD) method. Isolates were considered as susceptible when MICs of colistin were ≤2 mg/L and resistant when MICs were >2 mg/L (http://www.eucast.org/clinical_breakpoints/, accessed on 25 December 2020).

The RapidResa Polymyxin *Acinetobacter* NP^®^ assay was performed according to the manufacturer‘s recommendations from fresh overnight bacterial colonies grown at 37 °C on non-selective agar plates. UriSelect 4 agar plates (Bio-Rad, Marnes-la-Coquette, France) were used in this study for overnight cultures of *A. baumannii* isolates. Briefly, isolated colonies were suspended from an overnight agar plate into a vial of Mueller-Hinton II broth provided in the kit. The suspension was adjusted to obtain a turbidity equivalent to 0.5 McFarland standard. Therefore, the suspension contained approximatively 1.2 × 10^8^ CFU/mL that was used within 15 min. Then, 200 µL of Mueller-Hinton II broth were added to well a (as sterility control). Then, 200 µL of isolate suspension (tested isolate) were added in wells b, c (Figure 1). Then, 20 µL of the resazurin solution (provided in the kit) were added to each well in a single column. The reagent was mixed with the medium by pipetting up and down. The panels were then covered with the lid provided and incubated for up to 3–4 h in ambient air at 36 ± 1 °C and the color of the well inspected every 1 h (results are mostly obtained after 3 h of incubation). Colistin-resistant isolates grew in the absence and the presence of colistin (wells b and c) (i.e., purple or pink, indicating polymyxin resistance) while colistin-susceptible isolates grew only in wells b that did not contain any colistin (i.e., purple or pink) but not in wells c (i.e., blue, indicating polymyxin susceptibility) (Figure 1). Such a panel allows testing of eight different isolates.

## 3. Results and Discussion

Among the 62 *A. baumannii* isolates tested to evaluate the performance of the RapidResa Polymyxin *Acinetobacter* NP^®^ test, 36 were colistin susceptible (MICs of colistin ranging from <0.125 to 1 mg/L) and 26 isolates were resistant (MICs of colistin ranging from 4 to >128 mg/L), according to the results of the BMD method (Table 1). The 36 colistin-susceptible *A. baumannii* isolates gave negative results according to the RapidResa Polymyxin *Acinetobacter* NP^®^ test, except for a single isolate (MICs of 0.25 mg/L), which gave false-positive results (Table 1). Out of the 26 colistin-resistant *A. baumannii* strains, 25 strains were correctly identified as colistin resistant using the RapidResa Polymyxin *Acinetobacter* NP^®^ test. Only one colistin-resistant *A. baumannii* strain gave a false negative result. Additionally, six colistin heteroresistant isolates were also tested, which gave positive results (indicating colistin resistance). The specificity (which measures the proportion of negatives that are correctly identified) was calculated at 96%, and the sensitivity (which measures the proportion of positives that are correctly identified) was calculated at 97%.

In another study, Kon et al. evaluated the performance of another commercial kit (Rapid Polymyxin^TM^
*Acinetobacter* (RP-AB) (ELITechGroup, Puteaux, France) for colistin susceptibility testing [19]. The Rapid Polymyxin^TM^
*Acinetobacter* (ELITechGroup) is based on the colorimetric detection of glucose metabolization associated with bacterial growth resulting in a color change (red to yellow) of the pH indicator (phenol red) while the present test, the RapidResa Polymyxin *Acinetobacter* NP^®^ test (Liofilchem), is based on the reduction of the blue resazurin to the pink product resorufin by metabolically active bacteria cells. Unlike the RapidResa Polymyxin *Acinetobacter* NP^®^ test (Liofilchem), the results of RP-AB (ELITechGroup) were difficult to interpret and poorly correlated with the results of the BMD method, resulting in an unacceptable rate of very major errors. It showed a sensitivity and specificity of 41.2% and 86.1%, respectively [19]. Previous studies have evaluated the performance of the homemade test, the Rapid ResaPolymyxin NP test, using a collection of *A. baumannii* clinical isolates [16,17]. The sensitivity and specificity of the Rapid ResaPolymyxin NP test compared to the broth microdilution method was 100 and 96%, respectively, suggesting that this test is highly sensitive and specific for the detection of colistin resistance in *A. baumannii* [16]. In another study, the Rapid ResaPolymyxin NP test showed a 95.1% categorical agreement with results of the standard broth microdilution method [17].

Compared to Rapid Polymyxin^TM^
*Acinetobacter* (ELITechGroup), the RapidResa Polymyxin *Acinetobacter* NP^®^ test (Liofilchem) showed an excellent concordance with the results of the susceptibility test (BMD), for susceptible and resistant isolates.

## 4. Conclusions and Future Perspective

This study showed that the RapidResa Polymyxin *Acinetobacter* NP^®^ test (Liofilchem) is a reliable technique for detecting polymyxin resistance in *A. baumannii*. Further studies may further analyze the efficacy of this test to rapidly detect colistin resistance among additional clinically significant *Acinetobacter* spp. This test is rapid and easy to implement in any laboratory. It offers the possibility of detecting polymyxin resistance from bacterial cultures in 3–4 h. Although the BMD method requires an additional 18 h for obtaining the results, results of the RapidResa Polymyxin *Acinetobacter* NP^®^ can be obtained on the same day. Therefore, the use of this test may contribute to optimization of the antibiotic stewardship of infections due to multidrug- or pandrug-resistant *A. baumannii* strains that are of increasing frequency worldwide.

## Figures and Tables

**Figure 1 antibiotics-10-00558-f001:**
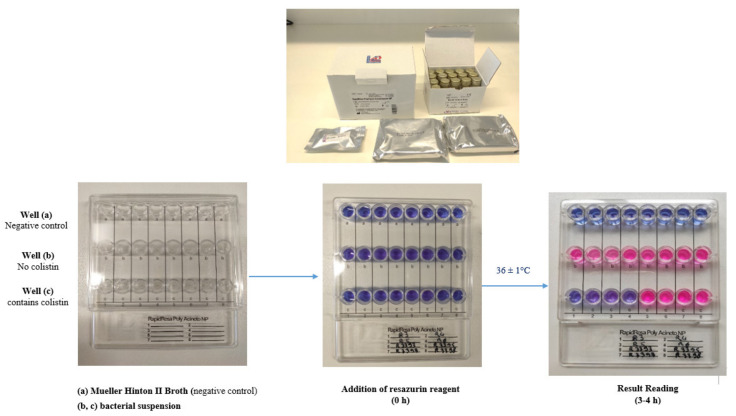
Representative results of the RapidResa Polymyxin *Acinetobacter* NP^®^ test. Non-inoculated wells are shown as negative controls for the medium and the color change (first row, a). The RapidResa Polymyxin *Acinetobacter* NP^®^ test was performed with colistin-susceptible isolates (1, 2, 3, 4) and with colistin-resistant isolates (5, 6, 7, 8) in a reaction without (second row, b) and with (third row, c) colistin. Bacterial growth is evidenced by color change of the culture medium from blue to pink or purple.

**Table 1 antibiotics-10-00558-t001:** RapidResa Polymyxin *Acinetobacter* NP^®^ test for rapid detection of polymyxin resistance in *A. baumannii* isolates.

Strain	Origin		Phenotype ^a^	BMD MIC Colistin (mg/L)	RapidResa Polymyxin *Acinetobacter* NP^®^ Test	
					Result	Discrepancy with BMD MIC Colistin Result ^b^
R3	France	Clinical	S	<0.125	Negative	-
R4	France	Clinical	S	<0.125	Negative	-
R5	France	Clinical	S	<0.125	Negative	-
R6	France	Clinical	S	<0.125	Negative	-
R8	France	Clinical	S	<0.125	Negative	-
R11	France	Clinical	S	<0.125	Negative	-
R14	France	Clinical	S	<0.125	Negative	-
R16	France	Clinical	S	<0.125	Negative	-
R21	France	Clinical	S	<0.125	Negative	-
R27	France	Clinical	S	<0.125	Negative	-
R32	France	Clinical	S	<0.125	Negative	-
R34	France	Clinical	S	<0.125	Negative	-
R35	France	Environmental	S	<0.125	Negative	-
R2536	Portugal	Clinical	S	0.5	Negative	-
R3381	Switzerland	Clinical	S	<0.125	Negative	-
R3382	Turkey	Clinical	S	<0.125	Negative	-
R3383	Turkey	Clinical	S	<0.125	Negative	-
R3384	Turkey	Clinical	S	<0.125	Negative	-
R3385	Turkey	Clinical	S	<0.125	Negative	-
R3386	Turkey	Clinical	S	<0.125	Negative	-
R3387	Turkey	Clinical	S	<0.125	Negative	-
R3389	Switzerland	Clinical	S	<0.125	Negative	-
R3391	Switzerland	Clinical	S	<0.125	Negative	-
R4319	Switzerland	Clinical	S	<0.125	Negative	-
R4321	Switzerland	Clinical	S	<0.125	Negative	-
R4323	Switzerland	Clinical	S	1	Negative	-
R4685	Lybia	Clinical	S	<0.125	Negative	-
R4687	Lybia	Clinical	S	<0.125	Negative	-
R4692	Lybia	Clinical	S	<0.125	Negative	-
R1401	Switzerland	Clinical	S	<0.125	Negative	-
R854	Greece	Clinical	S	<0.125	Negative	-
N581	Switzerland	Clinical	S	<0.125	Negative	-
N891	Switzerland	Clinical	S	<0.125	Negative	-
N211	Switzerland	Clinical	S	<0.125	Negative	-
N223	Switzerland	Clinical	S	<0.125	Negative	-
N3390	Switzerland	Clinical	S	0.25	Positive	Yes
R3393	Italy	Clinical	R	8	Positive	-
R3395	Italy	Clinical	R	64	Positive	-
R3396	Spain	Clinical	R	4	Positive	-
R3397	Spain	Clinical	R	16	Positive	-
R3398	Switzerland	Clinical	R	128	Positive	-
R3399	Turkey	Clinical	R	16	Negative	Yes
R3400	Turkey	Clinical	R	8	Positive	-
R3401	Turkey	Clinical	R	32	Positive	-
R3402	Turkey	Clinical	R	32	Positive	-
R3403	Turkey	Clinical	R	>128	Positive	-
R3404	Turkey	Clinical	R	4	Positive	-
R3405	Turkey	Clinical	R	>128	Positive	-
R3557	Nigeria	Environmental	R	32	Positive	-
R3804	Czech Republic	Veterinary	R	16	Positive	-
R3811	Switzerland	Clinical	R	16	Positive	-
R4322	Switzerland	Clinical	R	>128	Positive	-
R4326	Switzerland	Clinical	R	16	Positive	-
R4327	Switzerland	Clinical	R	128	Positive	-
R4330	Switzerland	Clinical	R	>128	Positive	-
R4684	Lybia	Clinical	R	128	Positive	-
R4686	Lybia	Clinical	R	8	Positive	-
R4320	Switzerland	Clinical	R	>128	Positive	-
N1432	Switzerland	Clinical	R	64	Positive	-
N1472	Switzerland	Clinical	R	16	Positive	-
N685	Switzerland	Clinical	R	4	Positive	-
N690	Switzerland	Clinical	R	64	Positive	-

^a^ S, susceptible; R, resistant. ^b^ (-) no discrepancy, BMD, broth microdilution. EUCAST clinical breakpoint for colistin: S ≤ 2, R > 2.

## Data Availability

Data is contained within the article.

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
