# Peer review of "RapidResa Polymyxin Acinetobacter NP® Test for Rapid Detection of Polymyxin Resistance in Acinetobacter baumannii"

_antibiotics, 2021, doi:10.3390/antibiotics10050558_

Round 1
Reviewer 1 Report
Authors proposed a brief report entitled “ RapidResa Polymyxin Acinetobacter NP® Test for rapid detec-2 tion of polymyxin resistance in Acinetobacter baumannii” for publication in Antibiotics, MDPI.
The main results of this paper are mainly summarized in Table 1.
Even if it is a brief report, I think that Authors could improve this paper by developing the study of the state of art. I understand that it is just a report paper, but only 7 references are cited; I think it is not sufficient.
I have some other issues:
Line 13. check the format of this email address. it should be smaller.
Line 19. Please, be more clear about this sentence: “The sensitivity and the spec-19 ificity of the test were found to be 96% and 97%, respectively.”
Line 30. “acquire resistance to virtually all antibiotics”. I would say “to all antibiotics, virtually”.
In my opinion, the introduction could be expanded with a deeper description of the state of the art.
moreover, this sentence about the scope of this paper could be developed “The aim of our present study was to evaluate this novel commercial test using a collection of polymyxin-susceptible and polymyxin-resistant A. baumannii isolates.”.
Line 68. “(http://www.eu-68 cast.org/clinical_breakpoints/).” this link is inserted in another format with respect to the text manuscript.
Line 99. As in the abstract, this sentence “the sensitivity and specificity of the 99 RapidResa Polymyxin Acinetobacter NP® test were found at 96% and 97%, respectively.” should be better explained.
Table 1. the columns Species A. baumannii could be eliminated from this table and just indicated in the caption of the table. It is not a varying information.
Results could be a separate section. Conclusions and future perspective could be the final paragraph.
Thank you
Author Response
Reviewer 1
Authors proposed a brief report entitled “RapidResa Polymyxin Acinetobacter NP® Test for rapid detection of polymyxin resistance in Acinetobacter baumannii” for publication in Antibiotics, MDPI.
The main results of this paper are mainly summarized in Table 1.
Even if it is a brief report, I think that Authors could improve this paper by developing the study of the state of art. I understand that it is just a report paper, but only 7 references are cited; I think it is not sufficient.
R. More references were added now.
I have some other issues:
Line 13. check the format of this email address. it should be smaller.
R. The format was checked.
Line 19. Please, be more clear about this sentence: “The sensitivity and the specificity of the test were found to be 96% and 97%, respectively.”
R. It is clearer now. All the colistin-susceptible A. baumannii isolates gave negative results according to the RapidResa Polymyxin Acinetobacter NP® test, except for a single isolate which gave false-positive result. Out of the 26 colistin-resistant A. baumannii strains, 25 strains were correctly identified as colistin resistant using the RapidResa Polymyxin Acinetobacter NP® test. Only one colistin-resistant A. baumannii strain gave a false negative result. Additionally, the six colistin heteroresistant isolates tested gave positive results. Altogether, the sensitivity and the specificity of the test were found to be 96% and 97%, respectively.
Line 30. “acquire resistance to virtually all antibiotics”. I would say “to all antibiotics, virtually”.
R. Modified as suggested by the reviewer.
In my opinion, the introduction could be expanded with a deeper description of the state of the art.
R. The introduction was expanded as suggested by the reviewer and more references were added.
moreover, this sentence about the scope of this paper could be developed “The aim of our present study was to evaluate this novel commercial test using a collection of polymyxin-susceptible and polymyxin-resistant A. baumannii isolates.”.
R. It was improved as suggested by the reviewer.
Line 68. “(http://www.eucast.org/clinical_breakpoints/).” this link is inserted in another format with respect to the text manuscript.
R. The link was inserted correctly.
Line 99. As in the abstract, this sentence “the sensitivity and specificity of the RapidResa Polymyxin Acinetobacter NP® test were found at 96% and 97%, respectively.” should be better explained.
R. The specificity (which measures the proportion of true negatives that are correctly identified) was calculated at 96 %, and the sensitivity (which measures the proportion of true positives that are correctly identified) was calculated at 97 %.
Table 1. the columns Species A. baumannii could be eliminated from this table and just indicated in the caption of the table. It is not a varying information.
R. The table was modified as suggested by the reviewer.
Results could be a separate section. Conclusions and future perspective could be the final paragraph.
R. Modified as suggested by the reviewer.
Reviewer 2 Report
In the manuscrit authors evalute a Rapid ResaPolymyxin Acinetobacter NP test, that allows categorization between colistin-susceptible/-resistant A. baumannii isolates in 4 h. It is based on the utilization of resazurin, because metabolically-active cells reduce the blue resazurin to the pink product resorufin. This method is visual and does not require specialized equipment.
This method was described by the last author in 2019, but has subsequently been evaluated by other authors in a manner similar to that presented in this manuscript. Jia et al, evaluated the test with 253 Gram - isolates, 58 of them A. baumannii (Evaluation of resazurin-based assay for rapid detection of polymyxin-resistant gram-negative bacteria. Jia H, Fang R, Lin J, Tian X, Zhao Y, Chen L, Cao J, Zhou T. BMC Microbiol. 2020 Jan 8;20(1):7. doi: 10.1186/s12866-019-1692-3.) and Germ J et al, evaluated 82 A. baumannii strains (Evaluation of resazurin-based rapid test to detect colistin resistance in Acinetobacter baumannii isolates. Germ J, Poirel L, Kisek TC, Spik VC, Seme K, Premru MM, Zupanc TL, Nordmann P, Pirs M. Eur J Clin Microbiol Infect Dis. 2019 Nov;38(11):2159-2162. doi: 10.1007/s10096-019-03657-1. Epub 2019 Aug 1). The authors do not cite these recent papers.
There are other Polymyxin Acinetobacter NP test (Performance of Rapid Polymyxin™ NP and Rapid Polymyxin™ Acinetobacter for the detection of polymyxin resistance in carbapenem-resistant Acinetobacter baumannii and Enterobacterales. Kon H, Abramov S, Amar Ben Dalak M, Elmaliach N, Schwartz D, Carmeli Y, Lellouche J. J Antimicrob Chemother. 2020 Jun 1;75(6):1484-1490. doi: 10.1093/jac/dkaa050) or (Rapid detection of colistin resistance in Acinetobacter baumannii using MALDI-TOF-based lipidomics on intact bacteria. Dortet L, Potron A, Bonnin RA, Plesiat P, Naas T, Filloux A, Larrouy-Maumus G. Sci Rep. 2018 Nov 15;8(1):16910. doi: 10.1038/s41598-018-35041-y.). The authors could have cited them and compared their results with this others method.
Author Response
Comments and Suggestions for Authors
In the manuscript authors evaluate a Rapid ResaPolymyxin Acinetobacter NP test, that allows categorization between colistin-susceptible/-resistant A. baumannii isolates in 4 h. It is based on the utilization of resazurin, because metabolically-active cells reduce the blue resazurin to the pink product resorufin. This method is visual and does not require specialized equipment. This method was described by the last author in 2019, but has subsequently been evaluated by other authors in a manner similar to that presented in this manuscript.
Jia et al, evaluated the test with 253 Gram - isolates, 58 of them A. baumannii (Evaluation of resazurin-based assay for rapid detection of polymyxin-resistant gram-negative bacteria. Jia H, Fang R, Lin J, Tian X, Zhao Y, Chen L, Cao J, Zhou T. BMC Microbiol. 2020 Jan 8;20(1):7. doi: 10.1186/s12866-019-1692-3.) and Germ J et al, evaluated 82 A. baumannii strains (Evaluation of resazurin-based rapid test to detect colistin resistance in Acinetobacter baumannii isolates. Germ J, Poirel L, Kisek TC, Spik VC, Seme K, Premru MM, Zupanc TL, Nordmann P, Pirs M. Eur J Clin Microbiol Infect Dis. 2019 Nov;38(11):2159-2162. doi: 10.1007/s10096-019-03657-1. Epub 2019 Aug 1). The authors do not cite these recent papers. There is other Polymyxin Acinetobacter NP test (Performance of Rapid Polymyxin™ NP and Rapid Polymyxin™ Acinetobacter for the detection of polymyxin resistance in carbapenem-resistant Acinetobacter baumannii and Enterobacterales. Kon H, Abramov S, Amar Ben Dalak M, Elmaliach N, Schwartz D, Carmeli Y, Lellouche J. J Antimicrob Chemother. 2020 Jun 1;75(6):1484-1490. doi: 10.1093/jac/dkaa050) or (Rapid detection of colistin resistance in Acinetobacter baumannii using MALDI-TOF-based lipidomics on intact bacteria. Dortet L, Potron A, Bonnin RA, Plesiat P, Naas T, Filloux A, Larrouy-Maumus G. Sci Rep. 2018 Nov 15;8(1):16910. doi: 10.1038/s41598-018-35041-y.). The authors could have cited them and compared their results with this others method.
R. Done as suggested by the reviewer.
Reviewer 3 Report
Rapid methodes for colistin susceptibility testing are highly desirable in the clinical microbiology laboratory. Professor Nordmann developed a valuable method to determine colistin resistance, based on resazurin oxidation, which can be applied to glucose non fermentin Gram negative species. In this work the commercial version of the method (RapidResa Polymyxin Acinetobacter NP® test) was proficiency-tested on a defined collection of colistin susceptible and resistant Acinetobacter baumannii isolates, providing substantially concordant results with the gold standard (broth microdilution).
MAIN REMARK
Fig. 1, mentioned on line 87, is not present in the manuscript and must be provided with the manuscript (I imagine it shows the colour change before and after 3-4 h incubation)
MINOR REMARKS:
On line 67, please spell "broth microdilution (BMD) method"
On line 109, please spell "BMD method" instead of "BMD technique"
Author Response
Rapid methods for colistin susceptibility testing are highly desirable in the clinical microbiology laboratory. Our lab has previously developed a valuable method to determine colistin resistance, based on resazurin oxidation, which can be applied to glucose non fermentin Gram negative species. In this work the commercial version of the method (RapidResa Polymyxin Acinetobacter NP® test) was proficiency-tested on a defined collection of colistin susceptible and resistant Acinetobacter baumannii isolates, providing substantially concordant results with the gold standard (broth microdilution).
MAIN REMARK
Fig. 1, mentioned on line 87, is not present in the manuscript and must be provided with the manuscript (I imagine it shows the colour change before and after 3-4 h incubation)
- Totally agreed. Figure was included as suggested by the reviewer.
MINOR REMARKS:
On line 67, please spell "broth microdilution (BMD) method"
- Done as suggested by the reviewer.
On line 109, please spell "BMD method" instead of "BMD technique"
- Done as suggested by the reviewer.
Round 2
Reviewer 2 Report
In line 99 authors mentioned “(Figure 1)” but is not present
The authors indicate in line 67 “This test complements the Rapid Polymyxin NP® test, which works with Enterobacterales but is not appropriate for non-fermenters such as A. baumannii” but there is other Polymyxin Acinetobacter NP test (Performance of Rapid Polymyxin™ NP and Rapid Polymyxin™ Acinetobacter for the detection of polymyxin resistance in carbapenem-resistant Acinetobacter baumannii and Enterobacterales. Kon H, Abramov S, Amar Ben Dalak M, Elmaliach N, Schwartz D, Carmeli Y, Lellouche J. J Antimicrob Chemother. 2020 Jun 1;75(6):1484-1490. doi: 10.1093/jac/dkaa050). Kon et al demonstrate: “Of the Acinetobacter baumannii carbapenem-resistant isolates, 58.6% (51/87) were resistant to colistin by BMD.
Categorical agreement between RP-AB and BMD was 59.8% (52/87), major errors 13.9% (5/36) and very major errors 58.8%(30/51). Sensitivity of RP-AB was 41.2% and specificity was 86.1%.
Jia et al, evaluated the test with 253 Gram - isolates, 58 of them A. baumannii (Evaluation of resazurin-based assay for rapid detection of polymyxin-resistant gram-negative bacteria. Jia H, Fang R, Lin J, Tian X, Zhao Y, Chen L, Cao J, Zhou T. BMC Microbiol. 2020 Jan 8;20(1):7. doi: 10.1186/s12866-019-1692-3.) and Germ J et al, evaluated 82 A. baumannii strains (Evaluation of resazurin-based rapid test to detect colistin resistance in Acinetobacter baumannii isolates. Germ J, Poirel L, Kisek TC, Spik VC, Seme K, Premru MM, Zupanc TL, Nordmann P, Pirs M. Eur J Clin Microbiol Infect Dis. 2019 Nov;38(11):2159-2162. doi: 10.1007/s10096-019-03657-1. Epub 2019 Aug 1). In my opinion this article does not add much novelty to the previously cited articles. The authors do not comment on these previous results in the discussion and do not compare them with their own results.
Author Response
The industrial version of SuperCAZ/AVI® medium developed for screening CAZ/AVI resistant Gram-negative isolates has been evaluated here using a collection of 87 well-characterized clinical isolates of worldwide origin. In addition, testing was performed by spiking stools with a series of resistant and susceptible isolates. In those conditions, the SuperCAZ/AVI®medium exhibited a sensitivity and specificity of 100 %, down to the lower limit of detection of 101 to 102 CFU/ml. The SuperCAZ/AVI® medium is a sensitive and specific screening medium for detection of CZA-resistant bacteria regardless of their resistance mechanisms.
Round 3
Reviewer 2 Report
The authors have correctly replied to the comments